# Spying on SARS-CoV-2 with Fluorescent Tags and Protease Reporters

**DOI:** 10.3390/v15102005

**Published:** 2023-09-27

**Authors:** Tsz-Leung To, Xiaoquan Li, Xiaokun Shu

**Affiliations:** 1Broad Institute, Cambridge, MA 02142, USA; 2Department of Pharmaceutical Chemistry, Cardiovascular Research Institute, UC San Francisco, San Francisco, CA 94158, USA

**Keywords:** SARS-CoV-2, fluorescence microscopy, fluorescent protein, genetically encoded reporter, viral protease, virus life cycle, drug screening, diagnostics

## Abstract

The SARS-CoV-2 coronavirus has caused worldwide disruption through the COVID-19 pandemic, providing a sobering reminder of the profound impact viruses can have on human well-being. Understanding virus life cycles and interactions with host cells lays the groundwork for exploring therapeutic strategies against virus-related diseases. Fluorescence microscopy plays a vital role in virus imaging, offering high spatiotemporal resolution, sensitivity, and spectroscopic versatility. In this opinion piece, we first highlight two recent techniques, SunTag and StayGold, for the in situ imaging of viral RNA translation and viral assembly. Next, we discuss a new class of genetically encoded fluorogenic protease reporters, such as FlipGFP, which can be customized to monitor SARS-CoV-2’s main (M^pro^) or papain-like (PL^pro^) protease activity. These assays have proven effective in identifying potential antivirals through high-throughput screening, making fluorogenic viral protease reporters a promising platform for viral disease diagnostics and therapeutics.

## 1. Introduction

Viruses, as intracellular parasites, can induce diseases by commandeering the host’s cellular machinery to complete their life cycle. They typically consist of genetic material (RNA or DNA), structural proteins (e.g., nucleocapsid), functional proteins (e.g., RNA-dependent RNA polymerase, RdRp), and lipid membranes (found in enveloped viruses). Upon entering a cell, the virus’s genetic material is released and either directly translated or transcribed and then translated into structural and functional viral proteins. Subsequently, these viral components assemble within the infected cell before being released into extracellular spaces to initiate another viral life cycle. The methods used by different viruses to infiltrate cells, manipulate host machinery, and exit cells vary based on the interactions between viral components and cellular structures. Understanding these intricate biological processes is crucial for the development of treatments against virus-induced diseases.

Early research on viruses, given their small size, predominantly relied on transmission electron microscopy (TEM) and biochemical assays to study viral infections. TEM, a classic technique, unveiled the exquisite structures of these minuscule viral particles, yielding valuable insights [1]. In vitro biochemical assays with bulk-prepared samples provided substantial knowledge about viral activities [2]. Recent advances in sequencing, mass spectrometry, and CRISPR-based screening [3,4,5] have further deepened our understanding of viral infections. Nevertheless, the dynamic aspects, which are at the core of these biological processes, remain largely elusive due to technical constraints. It is critical to directly and dynamically visualize infection processes within live cells to uncover the mechanisms driving infection and viral proliferation. Simultaneously, the time and labor-intensive nature of assessing potential drug candidates through traditional screening methods like plaque assays and qPCR underscore the urgent need for validated and rapid assays to evaluate viral infections.

Fluorescence imaging stands as a potent and adaptable modality, offering high spatial and temporal resolution, coupled with the advantages of sensitivity, spectroscopic flexibility, and amenability to high-throughput experimentation. It is a valuable tool for tracking molecular activity within cells. Notably, fluorescent protein (FP)-based biosensors, easily directed to various subcellular locations, have been instrumental in probing diverse biochemical processes, including protein–protein interactions, signaling events, and protease activation [6]. Over recent decades, advances in fluorescent imaging techniques and labeling approaches have greatly empowered the study of biological processes, notably viral infections. Timelapse imaging at the single viral particle level within live cells has yielded insights into dynamic infections and interactions with host machinery [7]. Viral protease activity reporters offer a direct and versatile means for large-scale screening and in vivo observations of viral infections. This opinion piece covers fluorescent tagging techniques like SunTag and super-photostable GFP StayGold for visualizing viral signals. Additionally, we explore a novel class of genetically encoded fluorogenic reporters for assessing viral infection through protease activity. Cell-based assays employing the fluorogenic protease reporter FlipGFP have been developed for detecting the activity of SARS-CoV-2’s main (M^pro^) or papain-like (PL^pro^) proteases, leading to the successful identification of potential antiviral candidates in high-throughput screenings.

## 2. Fluorescent Tagging Techniques for Visualizing Virus Life Cycle

Given the minuscule size of virus particles, signal amplification becomes critical when tagging single viral particles for fluorescent imaging. Protein multimerization achieves this, exemplified by the SunTag system, originally developed by Vale and colleagues [1]. It employs synthetic SunTag peptides that can recruit up to 24 copies of a protein. By fusing the SunTag to the N-terminus of a protein of interest (POI) and combining it with fluorescently labeled intracellular single-chain variable fragment-GFP (scFv-GFP), the POI becomes observable at the single-molecule level as translation begins for the SunTag-fused-POI (Figure 1). This principle extends to visualizing the translation of single viral RNAs (vRNAs) by fusing five SunTag peptide repeats to the N-terminus of the coxsackievirus B3 (CVB3) viral polyprotein (SunTag-CVB3) [2]. Thus, nascent polypeptides associated with translating vRNAs manifest as GFP foci in fluorescent microscopy. Timelapse imaging post virus infection unveiled remarkable heterogeneity in virus replication dynamics among cells, shedding light on extensive coordination between the translation and replication of single viral RNAs. In a related study, the same approach characterized the dynamic aspects of encephalomyocarditis virus (EMCV) infection and host cell antiviral signaling [3]. It identified distinct infection phases through SunTag imaging, demonstrating variations in the probability of antiviral response activation during infection. Notably, this response was more likely in cells with low viral replication rates in the initial hours of infection, and differences in antiviral responses among infected cells correlated with the strength of the dsRNA-sensing pathway. Such dynamic observations would be unattainable without single virus imaging.

The advent of super-resolution microscopy, surpassing the limitations of traditional light microscopy, offers a powerful tool for investigating small viruses. Its application in viral research has yielded unprecedented insights into virus behavior within live cells. Nevertheless, the constant illumination required by super-resolution microscopy has hindered its widespread use due to the photobleaching of traditional fluorophores like conventional GFP. The development of photostable fluorescent proteins (FPs) has been challenging, often resulting in reduced brightness, with few exceptions. Recently, an exceptionally photostable FP named StayGold was engineered from the jellyfish *C. uchidae* by Miyawaki and colleagues [4]. Unlike other photostable FPs, a single-point mutation substantially increased the brightness of StayGold without compromising its photostability. Using StayGold, researchers visualized SARS-CoV-2 spike proteins during infection using super-resolution microscopy (Figure 1). They characterized virus-induced endoplasmic reticulum (ER) membrane structures, including convoluted membranes (CMs) and double-membrane vesicles (DMVs), achieving resolutions previously attainable only with transmission electron microscopy. Furthermore, they conducted large-scale super-resolution microscopy observations, feasible only with a super-stable fluorescent protein, revealing intracellular trafficking and release processes.

## 3. Genetically Encoded Fluorogenic Protease Reporters for Viral Proteases

Proteases are vital enzymes in viral replication and maturation, and are integral to the life cycles of most viruses. The structural and functional proteins of many viruses are initially synthesized as polyproteins, requiring subsequent proteolytic splicing by viral-encoded proteases for functional maturation. Therefore, viral protease activity serves as a crucial drug target for developing antiviral therapeutics. These proteases are indispensable for processing viral polyproteins into individual functional proteins during the viral replication cycle. Detecting these protease activities offers significant advantages, enabling virus infection monitoring at the single-cell level and facilitating high-throughput screening for the identification of inhibitors against viral proteases.

Many previous genetically encoded protease reporters rely on Förster resonance energy transfer (FRET) rather than fluorescence [5,6]. FRET-based reporters often suffer from weak signals, necessitating labor-intensive image processing to derive ratiometric signals. To overcome these limitations, new protease reporters have been developed. These reporters modify their fluorescence signals in response to the protease of interest through one of the following mechanisms: (i) dimerization-dependent fluorescent protein exchange [7], (ii) protein splicing of the circularly permuted Venus fluorescent protein [8], or (iii) elimination of a peptide that oligomerizes and quenches GFP signals [9].

The Shu Lab has pioneered a range of novel multicolor fluorogenic protease reporters. These include the near-infrared reporter iCasper [10], the green GFP-based reporters ZipGFP [11] and FlipGFP [12], and the red mCherry-based reporter FlipCherry [12]. Although our prior focus was on investigating caspase activation and the dynamics of programmed cell death [13], these versatile protease reporters are modular and can be readily adapted to monitor the activity of viral proteases.

### 3.1. iProtease

The iProtease reporter arises from the redesign of a naturally monomeric near-infrared fluorescent protein (mIFP) [14], where chromophore incorporation hinges on protease activity regulation (Figure 2A). This innovation leverages the distinctive interaction between the chromophore biliverdin (BV) and mIFP, derived from bacterial phytochrome (BphP). iProtease delivers an extensive dynamic range, transitioning from darkness to brightness, with rapid kinetics, responding within seconds. By introducing a specific cleavage sequence of an executioner caspase, the resulting iCasper reporter has illuminated caspase-3 activity and apoptotic dynamics in cellular and animal contexts. In vivo iCasper imaging unveils the spatiotemporal link between apoptosis and embryonic morphogenesis, as well as the dynamics of apoptosis during brain tumorigenesis in Drosophila [10].

The modular design of the iProtease scaffold facilitates the engineering of reporters for various proteases by integrating distinct protease-specific cleavage sequences. Other research groups have harnessed the iProtease scaffold. Elledge and colleagues devised an iProtease-based granzyme B (GzB) activity reporter by introducing a GzB-specific cleavage sequence into the scaffold [15]. This GzB reporter contributed to a high-throughput platform for identifying candidate antigens recognized by T cells. Elowitz and colleagues employed the iProtease-based TEV reporter as a readout for circuit output in their synthetic protein-level pulse-generation circuit design [16]. iProtease has established a foundation for engineering fluorogenic reporters, including calcium, signaling molecules, and metabolites. For instance, Campbell and colleagues transformed mIFP into a near-infrared calcium reporter called NIR-GECO [17].

To expand the iProtease reporters with specific spectroscopic properties, we can incorporate various bacteriophytochrome-based far-red and near-infrared fluorescent proteins [18,19,20] into the iProtease design. The wealth of bacterial phytochromes available in protein sequence databases offers additional options for engineering fluorogenic reporters. Importantly, these near-infrared protease reporters will facilitate multiplex cell signaling imaging alongside green and red fluorescent kinase activity and protein–protein interaction reporters [21,22,23,24,25,26,27,28,29]. Additionally, these near-infrared reporters can be employed in conjunction with various chemogenetic tools [30,31,32] without spectral interference.

### 3.2. ZipGFP

A GFP-based fluorogenic protease reporter, named ZipGFP, has been devised by caging or ‘zipping’ together the two components of a self-assembling split GFP [33], thereby governing its self-assembly through protease activity (Figure 2B). Upon protease activation, the ZipGFP-based TEV (tobacco etch virus) protease reporter experiences a tenfold increase in fluorescence. This ZipGFP-based reporter effectively visualizes apoptosis in live zebrafish embryos, providing precise spatiotemporal resolution [11]. It holds promise for monitoring apoptosis during animal development and for crafting protease reporters extending beyond executioner caspases.

### 3.3. FlipGFP and FlipCherry

To amplify the dynamic range further, FlipGFP is strategically designed by flipping a beta strand within a tripartite split GFP [34] (Figure 2C). Following protease activation and cleavage, the beta strand is restored, leading to GFP reconstitution and a 100-fold surge in fluorescence in a FlipGFP-based TEV protease reporter. FlipGFP has a quantum yield of 0.66, a 2.6-fold improvement over ZipGFP. An executioner caspase reporter based on FlipGFP effectively visualizes apoptosis in live zebrafish embryos and apoptotic cells in the midgut of Drosophila [12]. The FlipGFP-based caspase reporter proves valuable for apoptosis monitoring during animal development and for developing reporters for a broader range of proteases beyond caspases. This design concept has also been applied to a red fluorescent protein, mCherry, to engineer a red fluorogenic protease FlipCherry [12].

**Figure 2 viruses-15-02005-f002:**
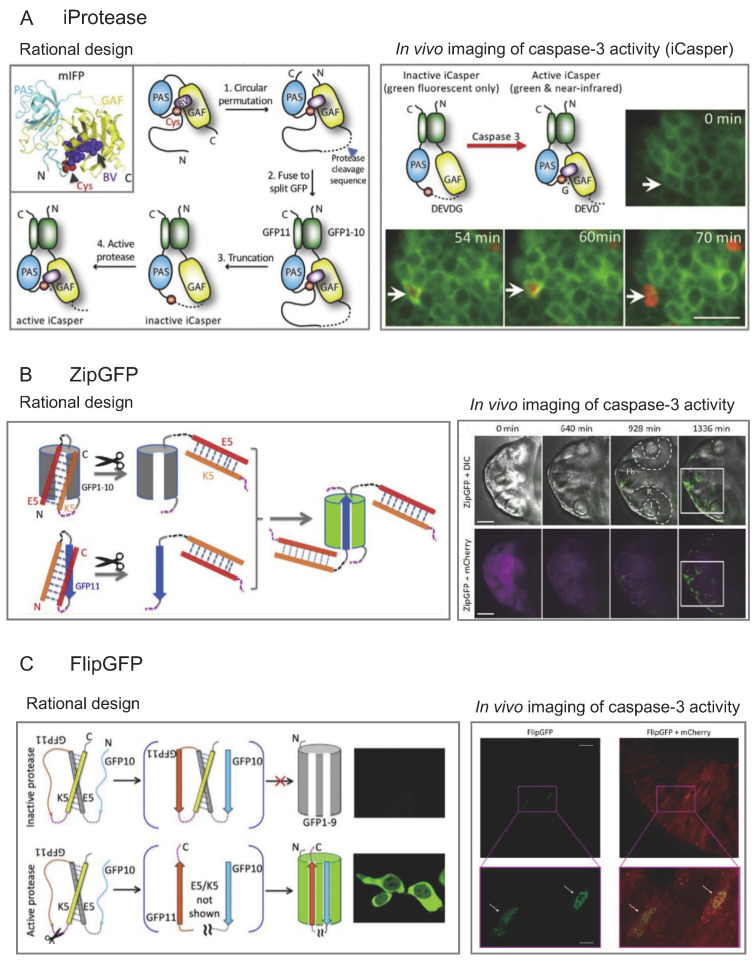
Multicolor fluorogenic protease reporters. (**A**) In the left panel, the near-infrared fluorogenic protease reporter, iProtease, is engineered from a naturally monomeric near-infrared fluorescent protein (mIFP) comprising a PAS domain followed by a GAF domain, derived from a truncated bacterial phytochrome (BphP). The primary structure of mIFP is depicted in the top left. A schematic diagram illustrates how mIFP is transformed into iProtease. It is known that in BphP, the chromophore biliverdin (BV) noncovalently binds to GAF and forms a thioether bond with a catalytic cysteine. The key principle behind iProtease is that the cleavage sequence connecting the catalytic cysteine to the C-terminus of GAF is shorter than the distance between the cysteine and the GAF domain. This displacement allows the cysteine to be released upon protease activation, returning to the BV-binding site, and restoring near-infrared fluorescence within seconds. In the right panel, iProtease, incorporating BV as the chromophore, visualizes caspase activity and apoptosis during the embryonic development of Drosophila (reprinted with permission from [10]. Copyright 2015 the National Academy of Sciences of the United States of America). (**B**) In the left panel, the bipartite GFP-based protease reporter, ZipGFP, is created by ‘zipping’ together the binding cavities of GFP1-10 and GFP11 with heterodimeric coiled coils E5 and K5, with the protease cleavage sequence highlighted in magenta. In the right panel, ZipGFP, featuring a caspase-3 cleavage sequence, allows the visualization of caspase activity and physiological apoptosis in zebrafish during normal embryo development following the gastrulation stage (reprinted with permission from [11]. Copyright 2016 Elsevier). (**C**) In the left panel, a tripartite GFP-based fluorogenic protease reporter, FlipGFP, is engineered by flipping a beta strand of GFP. A tripartite mCherry-based red fluorogenic protease reporter, FlipCherry, is also developed using the same principle. GFP1-9 comprises the fragment of GFP with the first nine beta strands and the central alpha helix containing the chromophore-forming amino acids. GFP10 and GFP11 represent the 10th and 11th beta strands of GFP, with GFP11 featuring the crucial Glu222 for chromophore maturation. In the right panel, FlipGFP, equipped with a caspase-3 cleavage sequence, visualizes caspase activity and physiological apoptosis in Drosophila enterocytes in the midgut (reprinted with permission from [12]. Copyright 2019 American Chemical Society).

## 4. FlipGFP-Based Assays for SARS-CoV-2’s Proteases

SARS-CoV-2 features two vital proteases: the main protease (M^pro^), also known as the chymotrypsin-like protease (3CL^pro^), and the papain-like protease (PL^pro^) [35]. These enzymes play a pivotal role in cleaving viral polypeptides at specific sites, yielding individual functional proteins crucial for viral replication and assembly (Figure 3A). Notably, PL^pro^ also contributes to modulating the host’s immune response [36]. Both M^pro^ and PL^pro^ have been investigated as potential therapeutic targets due to their potential to hinder the virus’s replication capabilities [37,38].

The modular design of the FlipGFP scaffold (Figure 3B) allows for the creation of protease reporters tailored to both the main and papain-like proteases of SARS-CoV-2 by integrating their respective protease-specific cleavage sequences (Figure 3C). This adaptable FlipGFP-based strategy has been adopted by multiple research teams. In an early study, Heaton and colleagues introduced a FlipGFP-based reporter system, emitting fluorescence post cleavage via M^pro^ [39]. This reporter was validated through evaluating the inhibition of SARS-CoV-2 M^pro^ using GC376, a known M^pro^ inhibitor, establishing a link between reporter signal and the suppression of SARS-CoV-2 replication. The resulting assay has facilitated high-throughput antiviral drug screening within a biosafety level 2 (BSL2) facility. A similar approach was undertaken by Tay and colleagues, employing a FlipGFP-based M^pro^ reporter to identify eight drugs, including the broad antiviral Masitinib, from a library of 1900 clinically safe drugs [40]. Concurrently, Wang and colleagues devised a cell-based FlipGFP assay for quantifying the potency of PL^pro^ inhibitors, also within a BSL2 environment [41]. Through the FlipGFP-PL^pro^ assay, two promising compounds were identified as effective inhibitors of SARS-CoV-2 replication in a cellular model. Finally, the Shu Lab and collaborators designed a FlipGFP-based M^pro^ reporter and conducted high-throughput screening in a BSL2 facility, screening an FDA-approved drug library [42]. Remarkably, one of the drugs identified through the FlipGFP assay, ethacridine, appears to primarily inactivate viral particles to exert its antiviral effect. Collectively, these studies underscore the potential of fluorogenic protease reporters for antiviral screening and assessment.

## 5. Conclusions

Recent advances in fluorescent imaging and labeling techniques have greatly facilitated the study of viral infections, significantly enhancing our understanding of the SARS-CoV-2 virus life cycle. Unlike electron microscopy or traditional assays that often require fixed or bulk samples, imaging single viral particles at the single-cell level provides deeper insights into the dynamic interactions between the virus and host cells during infection. This approach also illuminates the intricacies of viral infection dynamics. Moreover, fluorescence imaging streamlines high-throughput screening for potential antiviral compounds and offers a more mechanistic evaluation of viral infections compared to conventional methods. Given the critical role of the main and papain-like proteases in viral replication and assembly, monitoring their activities brings several advantages, including target specificity, early infection detection, streamlined drug discovery and development, real-time monitoring of viral activity, and exceptional signal-to-noise ratios.

Current imaging techniques predominantly take place in cell culture systems. The next step in this endeavor is to advance in vivo labeling and imaging methods for visualizing and characterizing virus infections in animal models. The fluorogenic protease reporters discussed in this opinion piece have already enabled the visualization of apoptotic signaling events in live animals with unprecedented spatial and temporal resolutions [10,11,12]. It is plausible that these fluorogenic protease reporters can be adapted for viral proteases in animal models, further enhancing our understanding of the spatiotemporal dynamics of viral infections and stimulating innovative approaches to diagnostics and therapeutics for not only SARS-CoV-2 but also other pathogenic viruses.

## Figures and Tables

**Figure 1 viruses-15-02005-f001:**
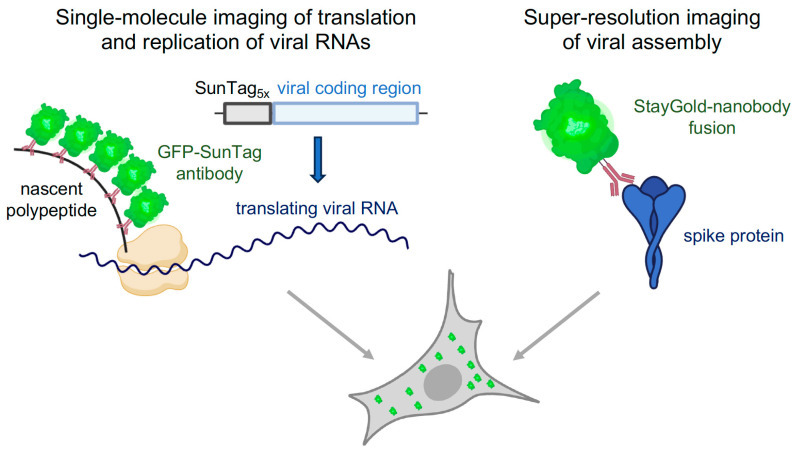
Innovative Fluorescent Tagging Techniques. In the left panel, the SunTag genetically fuses to the N-terminus of the viral genome region. This fusion enables the observation of nascent polypeptides associated with translating viral RNA at the single-molecule level, facilitated by an intracellular single-chain variable fragment-GFP fusion (scFv-GFP) targeting the SunTags [2]. Consequently, translating viral RNAs appear as distinct GFP foci under fluorescence microscopy. In the right panel, the exceptionally photostable fluorescent protein StayGold is fused with a high-affinity nanobody for the viral spike protein. This fusion allows the visualization of the SARS-CoV-2 assembly process by tracking the migration of SARS-CoV-2 spike proteins using StayGold under super-resolution microscopy [4].

**Figure 3 viruses-15-02005-f003:**
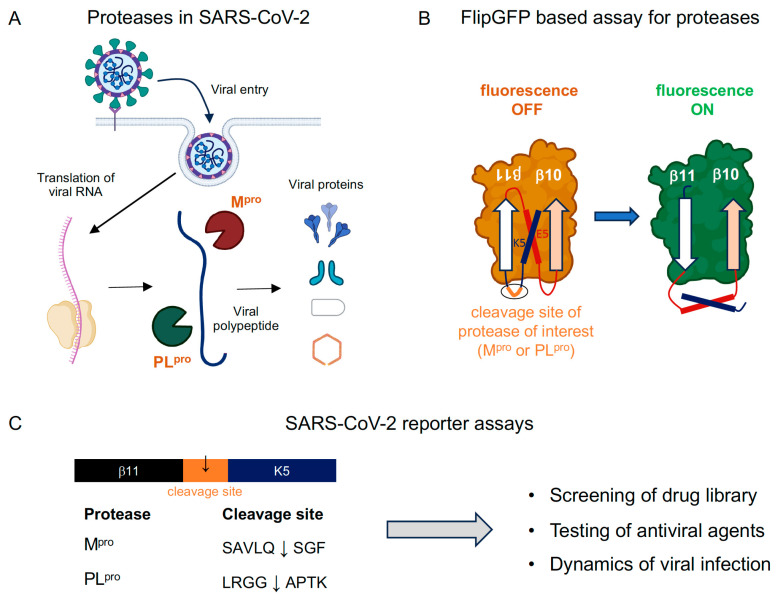
FlipGFP assays for SARS-CoV-2’s proteases. (**A**) In the SARS-CoV-2 virus life cycle, both the main protease (M^pro^) and the papain-like protease (PL^pro^) play pivotal roles in cleaving viral polypeptides at specific sites. This cleavage generates individual functional proteins essential for viral replication and assembly. (**B**) Incorporating the cleavage site of M^pro^ or PL^pro^ into the linker region between GFP11 and the K5 coil enables the creation of a protease activity reporter. (**C**) The cleavage sites for M^pro^ and PL^pro^, employed in published reports [39,40,41,42], have found diverse applications in the FlipGFP-based reporters.

## Data Availability

Further information and requests for resources and reagents should be directed to and will be fulfilled by Xiaokun Shu (xiaokun.shu@ucsf.edu).

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
