# Peer review of "Spying on SARS-CoV-2 with Fluorescent Tags and Protease Reporters"

_viruses, 2023, doi:10.3390/v15102005_

Round 1
Reviewer 1 Report
The manuscript “Spying on SARS-CoV-2 with Fluorescent Tags and Protease Reporters” is devoted to the “hot” problem of visualizing individual molecules and their transformations in a living cell using fluorescent technique.
Novel approaches to visualize viral translation and assembly in situ using SunTag and StayGold methods, and a new class of genetically-encoded fluorogenic protease reporters are presented. The authors explain in clear language the essence of the latest and interesting approaches to visualizing the stages of virus reproduction.
The introduction gives a concise but fairly complete idea of the features of virus research and the place of fluorescent research methods in their study.
Section 2 explains how individual molecules can be visualized during viral RNA replication using the SunTag system.
Recent developments in fluorescence imaging presented in the manuscript are impressive. Thus, possibility to visualize SARS-COVID19 spike proteins using super-resolution microscopy and a new photostable compound StayGold may lead to deeper understanding of the mechanisms of viral infection in cells.
The manuscript contains interesting material about proteases and their detection using genetically encoded protease reporters. Various fluorogenic protease reporters are analyzed and their use is demonstrated. The data on the studies of caspases, which led to the clarification of the mechanisms of apoptosis, look very relevant.
The authors provide an analysis of data on the use of protease reporters in relation to SARS-CoV-2 proteases and for the development of drugs against this virus. Viruses are intracellular parasites, realizing their life cycle at the level of intermolecular interactions. Undoubtedly, research at the level of individual molecules is the most justified approach to creating new antiviral drugs, and the results already obtained confirm this.
The “Spying on SARS-CoV-2 with Fluorescent Tags and Protease Reporters” manuscript is a compact, well-written overview of recent publications on the development and application of novel single-molecule imaging techniques using fluorescent tags. Such a review will be very useful for readers of Viruses.
Unfortunately, I was not able to understand why the article was classified in the “Opinions” section. In my opinion, it does not contain polemical materials.
Author Response
We deeply thank the reviewer for the positive and constructive feedback. We are particular grateful for their comment on how this piece will be very useful for the readers of Viruses.
To address this reviewer's question on why the article was classified in the “Opinions” section, rather than "Reviews", we need to point out the "Reviews" in Viruses are expected to cover their topics more comprehensively and in more depth. Our piece is not intended as an exhaustive review of fluorescence imaging of viruses, a topic that has been extensively covered elsewhere, including in a publication with PMID 34207305 from MDPI. Instead, it exclusively centers on selected new innovations in this field, aligning with the special topic of "Innovative Imaging in Viral Research." Therefore, we have decided to submit this piece as "Opinion".
Reviewer 2 Report
The manuscript offers a comprehensive overview of the recent methods employed to investigate the SARS-CoV-2 infection mechanism. It is insightful and compelling. While the manuscript is accepted as presented, I do have a query: the authors seem to have adapted figures from their prior works with slight alterations. Is this practice in line with copyright rules?"
Author Response
We deeply thank the reviewer for the positive and constructive feedback. We are particular grateful for their comment on the fact that they found this piece insightful and compelling.
To address this reviewer's inquiry about our obligation to adhere to copyright rules when adapting figures from our previous research in Figure 2, we have confirmed by reaching out to the National Academy of Sciences of the United States (NAS), Elsevier, and the American Chemical Society (ACS), respectively. In all three cases, we were granted either written permission (NAS and ACS) or a license number (5632501152176 from Elsevier) for reusing our prior work under this specific context. Therefore, there should be no copyright issues. We thank this reviewer for bringing up this important point.